

# The first report of the prevalence of *Nosema ceranae* in Bulgaria

Rositsa Shumkova[1], Ani Georgieva[2], Georgi Radoslavov[3,4], Daniela Sirakova[3], Gyulnas Dzhebir[4], Boyko Neov[3], Maria Bouga[5] and Peter Hristov[3,4]

[1] Agricultural and Stockbreeding Experimental Station, Agricultural Academy, Smolyan, Bulgaria
[2] Department of Pathology, Institute of Experimental Morphology, Pathology and Morphology and Anthropology with Museum, Bulgarian Academy of Sciences, Sofia, Bulgaria
[3] Department of Animal Diversity and Resources, Institute of Biodiversity and Ecosystem Research, Bulgarian Academy of Sciences, Sofia, Bulgaria
[4] Department of Structure and Function of Chromatin, Institute of Molecular Biology, Bulgarian Academy of Sciences, Sofia, Bulgaria
[5] Laboratory of Agricultural Zoology and Entomology, Agricultural University of Athens, Athens, Greece

## ABSTRACT

*Nosema apis* and *Nosema ceranae* are the two main microsporidian parasites causing nosematosis in the honey bee *Apis mellifera*. The aim of the present study is to investigate the presence of *Nosema apis* and *Nosema ceranae* in the area of Bulgaria. The 16S (SSU) rDNA gene region was chosen for analysis. A duplex PCR assay was performed on 108 honey bee samples from three different parts of the country (South, North and West Bulgaria). The results showed that the samples from the northern part of the country were with the highest prevalence (77.2%) for *Nosema ceranae* while those from the mountainous parts (the Rodopa Mountains, South Bulgaria) were with the lowest rate (13.9%). Infection with *Nosema apis* alone and co-infection *N. apis/N. ceranae* were not detected in any samples. These findings suggest that *Nosema ceranae* is the dominant species in the Bulgarian honey bee. It is not known when the introduction of *Nosema ceranae* in Bulgaria has occurred, but as in the rest of the world, this species has become the dominant one in Bulgarian *Apis mellifera*. In conclusion, this is the first report for molecular detection of *Nosema* infection of honey bee in Bulgaria. The results showed that *N. ceranae* is the main *Nosema* species in Bulgaria.

Corresponding author
Peter Hristov, peter_hristoff@abv.bg

## INTRODUCTION

The Western honey bee (*Apis mellifera* L., Hymenoptera: Apidae) is a species of crucial economic, agricultural and environmental importance. The biological significance of bees is rooted in the fact that they are main pollinators in the natural environment (*Barrios et al., 2016*; *Ballantyne et al., 2017*). About 80% of the pollination of entomophilous plants is carried out by *Apis mellifera*. In all crops, active pollination significantly increases their yields (*Partap, 2011*). On the other hand, honey bees are a valuable economic asset due to the ensemble of their products which includes honey, bee pollen, propolis, royal jelly, and bee venom, used by humans for food and treatment (*Pasupuleti et al., 2017*; *Sforcin, Bankova & Kuropatnicki, 2017*).

Bulgaria has long-standing traditions in the production of honey and bee products, a precondition for which is the varied and rich vegetation of the Balkan Peninsula suitable for the production of honey and also the favorable natural, climatic and ecological conditions.

Honey bee colonies suffer from numerous pathogens. These include various bacteria, viruses, fungi and endo- and ecto-parasites. Some of them, microsporidians, are obligate intracellular parasites belonging to the kingdom Fungi (*Keeling & McFadden, 1998*; *Hirt et al., 1999*; *Sina et al., 2005*). *Nosema* is a microsporidian genus causing an infection called Nosemosis of adult honey bees (*Klee et al., 2007*). Only two main species of *Nosema* causing infection in *Apis mellifera* have been recognized—*Nosema apis* (*Zander, 1909*) and *Nosema ceranae* (*Fries et al., 1996*). It is well known that *N. apis* is specific for the Western honey bee, *Apis mellifer* a L., whilst the Eastern honey bee *Apis cerana* harbours *Nosema ceranae* (*Fries et al., 1996*). However, many recent investigations have revealed that *N. ceranae* is not restricted only to *A. cerana*, but it transferred to *A. mellifera*, and even became a dominant species in many parts in the world (*Klee et al., 2007*; *Paxton et al., 2007*; *Chen et al., 2008*; *Invernizzi et al., 2009*; *Tapaszti et al., 2009*; *Stevanovic et al., 2011*; *Gajger et al., 2010*; *Ansari et al., 2017*; *Papini et al., 2017*). The exact time and transmission route of transfer of *N. ceranae* from *A. cerana* to *A. mellifera* is not known. It is possible that during the last decades, the rapid, long-distance dissemination of *N. ceranae* is likely due to the transport of infected honey bees and/or by the increased mobility of people, goods and livestock. Recently, a so called "Colony Collapse Disorder" (CCD) disease has been described in the United States (*Chen et al., 2008*) and Europe (*Topolska, Gajda & Hartwig, 2008*). The *Nosema ceranae* was suspected to one of the contributor to this illness, especially winter colony loses (*Klee et al., 2007*).

There are two main techniques for identifying *Nosema* species—microscopic and molecular. The microscopic methods such as Light microscopy, Giemsa and Toluidine staining, and Transmission electron microscopy were introduced first (*Fries et al., 2013*); but they are still a valuable, relatively cheap and simple method for screening and identification of *Nosema* infection. According to *Ptaszyńska et al. (2014)*, *N. ceranae* spores seem to be more sculptured with deeper ornamentation than those of *N. apis*. Despite the fact that *N. apis* and *N. ceranae* spores are morphologically different, in case of low rate of infection or presence of vegetative forms of *Nosema,* differentiation between spores of *N. apis* and *N. ceranae* is very difficult This requires the search for methods that are more sensitive. Therefore, various molecular methods have been developed. Those include mainly PCR techniques (conventional or duplex PCR, PCR-RFLP, qPCR) (*Fries et al., 2013*) involving usually a wide range of species-specific PCR primers (*Martín-Hernández et al., 2007*; *Klee et al., 2007*; *Chen et al., 2008*).

Until now, there has been no data regarding the distribution of *N. apis* and *N. ceranae* throughout Bulgaria as well as information if *N. ceranae* has become a dominant species, although Nosema infection for the surrounding Balkan countries is well studied (*Stevanovic et al., 2011*; *Whitaker, Szalanski & Kence, 2011*; *Hatjina et al., 2011*).

The main goal of the current study is to investigate and determine the presence and distribution of the two different *Nosema* spp. in the Bulgarian honey bee.

**Table 1  Distribution of *N. ceranae* in three different regions in Bulgaria.**

| Region | No. of collected samples | No. of *Nosema* positive samples | % of *Nosema* positive samples | *N. ceranae* | *N. apis* | Co-infection |
|---|---|---|---|---|---|---|
| Smolyan (SB) | 36 | 5 | 13.9 | 5 | – | – |
| Sofia (WB) | 28 | 18 | 64.3 | 18 | – | – |
| Russe (NB) | 44 | 34 | 77.2 | 34 | – | – |
| **Total** | **108** | **57** | **52.8** | **57** | **–** | **–** |

**Notes.**

SB, South Bulgaria; WB, West Bulgaria; NB, North Bulgaria.

# MATERIALS AND METHODS

## Sample collection

A total of 108 honey bee samples were collected from three different parts in the country: Rousse district (North Bulgaria, $N = 44$), Sofia district (West Bulgaria, $N = 28$) and Smolyan district (South Bulgaria, $N = 36$), (Table 1, Fig. 1) in April–May 2017. There is no bias concerning the obtained honey bee samples. The first two regions are characterized by their low-lying and generally flat plains, while the last region is situated in the Rodopa Mountains. Sampling was done according to the guidelines of the *Office International des Epizooties (2008)*. None of the honey bee colonies were treated against *Nosema* infection for at least six months. In each hive, five adult worker honey bees were randomly selected at the entrance of the hive or on frames away from the brood nest. The honey bees were placed in a falcon tube, put in a cooler bag and stored at −20 °C prior to analysis.

## DNA extraction

Briefly, prior to DNA extraction, the abdomen of a single bee was cut off with scissors, mechanically homogenized with a cell lysis buffer and centrifuged for 1 min at 15,000 rpm. Total DNA was isolated by using GeneMATRIX Tissue and Bacterial DNA purification Kit (Cat. No. E3551-01; EURx Ltd., Gdańsk, Poland) according to the manufacturer instructions. Shortly, the pellet was resuspended in a cell lysis buffer (a component of a DNA purification kit); proteinase K was added and incubated overnight at 56 °C. The extracted DNA was resuspended in 50 μL of elution buffer. The DNA concentration was determined spectrophotometrically and the quality of the DNA samples was examined on 1% agarose gel electrophoresis stained with Greensafe premium (Cat. No. MB13201; Nzytech, Lisbon, Portugal). The purified DNA was stored at −20 °C until PCR assay.

## Gene selection and PCR amplification

The small subunit (16S) ribosomal RNA gene was chosen for molecular identification of *Nosema ceranae* and *Nosema apis*. A fragment of this gene was amplified in both *Nosema* species using primers designed by *Martín-Hernández et al. (2007)*. 321APIS-FOR (5′-GGGGGCATGTCTTTGACGTACTATGTA-3′; 321APIS—REV (5′-GGGGGGCGTTTAAAATGGAAACAACTATG-3′) for Nosema apis and 218MITOC—FOR (5′-CGGCGACGATGTGATATGAAAATATTAA-3′); 218MITOC—REV (5′-CCCGGTCATTCTCAAACAAAAAACCG-3′) for *N. ceranae*.

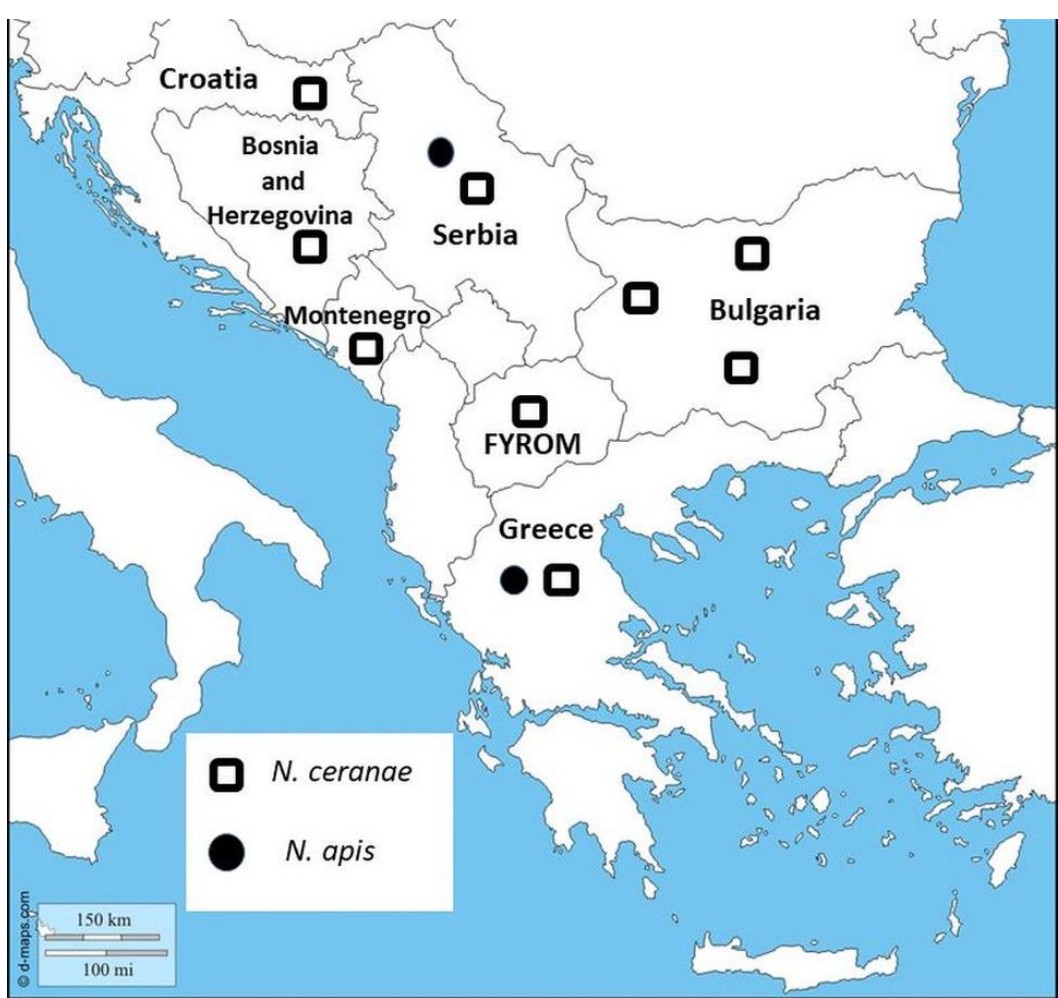

**Figure 1** **Map showing sampling locations in Bulgaria.** *Nosema* species distribution are represented in all Balkan countries.

Both primer sets were used together for performing a duplex PCR for identification and discrimination of *Nosema* species.

The expected number of the amplified products in *N. ceranae* using the 218MITOC primers can be either 218 or 219 depending on the sequences for *N. cerana* e available in the GenBank database (http://www.ncbi.nlm.nih.gov/) (*Martín-Hernández et al., 2007*). In the case of *N. apis*, the expected size of the amplicon using the 321APIS primers was 321 bp. In addition, a negative control was included for all PCR reactions. As a positive control, cytochrome c-oxidase gene (*CoI2*) of *Apis mellifera* was used in all studied samples. The sequence of primers used for positive control was CoI2-F (5′-CCTGATATAGCATTTCCTCG-3′) and CoI2-R (5′-TGTGAATGATCTAAAGGTGG-3′) designed on the base of the known mitochondrial genome of *A. m. ligustica* (Acc. No. L06178, *Crozier & Crozier, 1993*). The PCR mixtures contained 25 μL of NZYTaq 2× Colourless Master Mix (Cat. No. MB04002; Nzytech, Lisbon, Portugal), 0.4 μM of each

species-specific primer (FOR/REV), 1 µL of template DNA and PCR water (Cat. No. E0211-01; EURx Ltd., Gdańsk, Poland) in a total volume of 50 µL. All PCR reactions were carried out using a Little Genius thermocycler (BIOER Technology Co., Ltd) under the following conditions: initial denaturation at 94 °C for 5 min; 30 cycles (denaturation at 94 °C for 30 s; primer annealing at 50 °C for 30 s; extension at 72 °C for 1 min) and final extension at 72 °C for 10 min. PCR products were visualized on a 2% agarose gel with Greensafe premium (Cat. No. MB13201; Nzytech, Lisbon, Portugal). The fragment size was determined using Gene-Ruler$^{TM}$ 100 bp Ladder Plus (Cat. No. SM0323; ThermoFisher Scientific Inc., Waltham, MA, USA).

### Sequence analysis

The successfully amplified products for *Nosema* (20 samples) were purified by a PCR purification kit (Gene Matrix, PCR clean-up kit; EURx, Gdańsk, Poland) and sequenced in both directions by a PlateSeq kit (Eurofins Genomics, Ebersberg, Germany). The obtained sequences were deposited in GenBank under accession number MG657260.

## RESULTS

Duplex PCR with species-specific primers (321APIS-FOR/REV and 218MITOC-FOR/REV) produced PCR products in 57 samples out of 108 analyzed (52.8% successful amplifications), while 51 samples failed to produce a PCR product (47.2%). There were no PCR products in the negative controls. The results from the obtained sequences support identity only to *Nosema ceranae* species.

From all investigated samples only *Nosema ceranae* infection was detected. The highest level of infection was observed in North Bulgaria. From all 44 investigated samples, 34 (77.2%) were *Nosema* positive (Table 1). In the west part of the country (Sofia district), *Nosema* positive samples were detected in 18 from all 28 studied samples (64.3%). The lowest level of infections was found in the honey bee samples from the mountainous part of the country (Smolyan district, the Rodopa Mountains). From all 36 investigated samples, only five (13.9%) were *Nosema* positive. Surprisingly, in all the studied samples from three different regions of the country only *Nosema ceranae* was found. The presence of *Nosema apis* as well as co-infections *N. apis/N. ceranae* were not detected (Table 1). Moreover, the honey bee samples from the flat part of the country (Sofia and Rousse districts) had a higher prevalence of *N. ceranae* infection as compared with samples obtained from the mountainous part (Smolyan district).

## DISCUSSION

In the current study, we have presented for the first time molecular identification of two *Nosema* spp. and their distribution in Bulgaria. The results indicate the presence of *N. ceranae* in 57 of 108 investigated samples, which suggests the dominance of *N. ceranae* in all investigated regions. The results of many studies from the Balkan countries have indicated that *N. ceranae* displaces *N. apis* (*Stevanovic et al., 2011*; *Whitaker, Szalanski & Kence, 2011*; *Hatjina et al., 2011*; *Gajger et al., 2010*) (Fig. 1). One reasonable question is

why this introduced parasite (*N. ceranae*) has become in a short time the dominant species worldwide? Concerning the virulence of the two *Nosema* spp., the results are contradictory. It is an interesting fact that in many European countries numerous studies report that *N. ceranae* is more virulent and thus possesses a competitive advantage in compared to *N. apis* (*Klee et al., 2007*; *Paxton et al., 2007*; *Forsgren & Fries, 2010*). Contrary to this, more recent research, performed mainly in the USA, does not support these observations (*Huang et al., 2015*; *Milbrath et al., 2015*). These studies suggest that the US honey bees may be less susceptible to *N. ceranae* infections than European bees or that the US isolates of the pathogen are less infective and less virulent than European isolates.

These findings are a suitable way to explain our results. We found that *N. ceranae* infection prevailed in honey bee colonies from the flat part of the country (Rousse and Sofia district), while in the mountainous part (Smolyan district, the Rodopa Mountains) the prevalence was the lowest (Table 1). Different subspecies of *Apis mellifera* are raised in the flat and in mountainous regions. *A. m. macedonica* is considered to be a native honey bee for Bulgaria (*Ruttner, 1988*). More than three decades ago, *A. m. ligustica*, *A. m. carnica* and *A. m. caucasica* were introduced and were reared in Bulgaria (*Bouga et al., 2011*). These subspecies are disseminated mainly in the flat regions of the country. On the other hand, in Bulgaria exists a local honey bee subspecies called *A. m. rodopica*, geographically distributed only in the Rodopa Mountains massive (*Petrov, 1995*; *Bouga et al., 2011*; *Ivanova et al., 2012*; *Nikolova & Ivanova, 2012*). Our previous investigation on mitochondrial heredity have revealed that *A. m. rodopica* is a distinct subspecies concerning malfunction of COI protein in compared to other honey bee subspecies reread in Bulgaria (*Radoslavov et al., 2017*).

According to *Petrov (2010)*, *A. m. rodopica* possesses a lot of advantages in compared to the introduced subspecies—good adaptation to the specific local climatic conditions, resistance numerous diseases etc. These findings might explain the differences in the prevalence of *N. ceranae* in the different areas since different subspecies of bees are reared in each area. Another fact, which may explain the low rate of infection of *A. m. rodopica*, is the long geographical isolation of this subspecies. Moreover, the beekeepers in this region are encouraged to raise this local honey bee and even not to allow genetic introgression with other subspecies in Bulgaria.

Another fact which may explain the high rate of infestation with *N. ceranae* in honey bee colonies from the plain regions in compared to the mountainous regions of the country is the different climatic conditions in these places. The Rodopa Mountains climate is rather colder than the other two investigated regions. This determine the later development of honey bee colonies (May–August). Moreover, malfunction of COI protein in *A. m. rodopica* may associated with lower mitochondrial respiration metabolism, which may determinate and lower activity of honey bees. Many papers have discussed that warmer climatic conditions favored prevalence of *N. ceranae* (*Tapaszti et al., 2009*; *Stevanovic et al., 2011*), whereas *N. apis* remains more prevalent in colder climates (*Budge et al., 2010*; *Gisder et al., 2010*; *Natsopoulou et al., 2015*).

## CONCLUSION

This is the first report of the distribution of *N. ceranae* of honey bee colonies in Bulgaria. We found that *N. ceranae* is the dominant species in the Bulgarian honey bee. A local honey bee *A. m. rodopica* reared in the Rodopa Mountains seems to be more resistant in comparison to the introduced species. Because of this, local honey bees should be kept as a part of the genetic diversity and the related conservation activities.

### Funding

This work was supported by the National Science Fund of the Bulgarian Ministry of Education and Science, Sofia (grant number 06/10 17.12.2016). The funders had no role in study design, data collection and analysis, decision to publish, or preparation of the manuscript.

### Grant Disclosures

The following grant information was disclosed by the authors:
National Science Fund of the Bulgarian Ministry of Education and Science: 06/10 17.12.2016.

### Competing Interests

The authors declare there are no competing interests.

### Author Contributions

- Rositsa Shumkova conceived and designed the experiments, obtained the samples.
- Ani Georgieva wrote the paper, reviewed drafts of the paper.
- Georgi Radoslavov conceived and designed the experiments, analyzed the data, contributed reagents/materials/analysis tools, wrote the paper, prepared figures and/or tables, reviewed drafts of the paper.
- Daniela Sirakova and Boyko Neov conceived and designed the experiments, performed the experiments, wrote the paper.
- Gyulnas Dzhebir performed the experiments, wrote the paper.
- Maria Bouga analyzed the data, wrote the paper, reviewed drafts of the paper.
- Peter Hristov conceived and designed the experiments, analyzed the data, contributed reagents/materials/analysis tools, wrote the paper, prepared figures and/or tables, pay APC.

### Data Availability

GenBank: MG657260.

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
