# Peer review of "The first report of the prevalence of Nosema ceranae in Bulgaria"

_PeerJ, doi:10.7717/peerj.4252_

## Round 0.1 · original submission · Major Revisions

· Academic Editor

Major Revisions

The manuscript represents a survey for nosematosis in Apis mellifera from Bulgaria where the authors detected only the presence of Nosema ceraneae while failed to find N. apis.

Even if the data presented here are not novel, I think that they are valuable.

Nevertheless, the article needs some major revisions prior to publication as it lacks some important details mainly involving the experimental design (methods do not seem to be described with sufficient information). In addition, the discussion needs to be more elaborated and conclusions drawn by the authors do not seem wholly appropriate.

Reviewer 1 ·

Basic reporting

In this study, Hristov et al. aimed to analyze the presence of Nosema apis and Nosema ceranae in samples taken from different regions of Bulgaria. The information provided is interesting. As it is mentioned in the General comments, I recommend performing some analysis in order to complement the manuscript.
The structure of the article is acceptable. I recommend some modifications in the introduction and to remove the figure.

Experimental design

More details of the experimental design should be provided.
L85-87- Which was the criteria of the number of samples taken from each region? Was it according with the number of colonies from each region? Please, provide details of the experimental design.
L91-92- Why were the samples taken from the entrance or from the frames away from the brood nest? Was any bias in the method of sample collection according with the region?
L95- Which cell lysis buffer? Please, provide information and references.
L125- Did the authors include positive controls? Did the authors verify the identity of at least one of the PCR fragments of each Nosema species? According to Martín-Hernández et al., 2007 the annealing of these primers is 61.8ºC and not 50ºC.

Validity of the findings

The findings are not novel but are valid. The conclusion should be revised.
L37- Please, remove “that confirms the worldwide dissemination and prevalence of Nosema ceranae”. The main conclusion is that N. ceranae is the main Nosema spp. in Bulgaria. As there are regions of the world where there is no information on the prevalence and distribution of this pathogen, it is not possible to say that these results confirm the worldwide dissemination. Also, I do not understand what the authors mean with that “confirms the prevalence of Nosema ceranae”.
L186-191- This information is not the conclusion. Please, remove it.

Comments for the author

L25- Please, correct “the objective of the present study was to”
L28- I think that the authors should re-organize the information, mentioning first that only N. ceranae was detected and not N. apis and then the percentage of infection from each region.
L29-30- “rate of infection”, not of invasion
L37- Please, remove “that confirms the worldwide dissemination and prevalence of Nosema ceranae”. The main conclusion is that N. ceranae is the main Nosema spp. in Bulgaria. As there are regions of the world where there is no information on the prevalence and distribution of this pathogen, it is not possible to say that these results confirm the worldwide dissemination. Also, I do not understand what the authors mean with that “confirms the prevalence of Nosema ceranae”.
L42-48- Please, remove this paragraph. This is a general background of the beekeeping activity in Bulgaria and it is not directly related to the purpose of the manuscript. I recommend starting the introduction section highlighting the importance of bees in pollination and for the ecosystem maintenance and then introduce the sanitary problems that the bees have to face. Authors can also mention that there have been reports of colony losses in different countries and that Nosema is one of the main threats that affect honeybees.
L49- rewrite these sentences in order to not repeat the words
L53- Please, provide references. Also, replace Nosematosis for Nosemosis and correct this in the whole manuscript.
L53-54- Nosema apis and Nosema ceranae are the only two microsporidia so far reported to infect honeybees. However, there are other Nosema spp. Please, correct this sentence and provide references.
L68-70- The difference in morphology between both Nosema species is only distinguishable by very expertise technicians. Authors can mention the morphological difference between both types of spores and the advantages of using molecular techniques to distinguish between both species.
L72- Evans et al., 2013 reviewed different techniques to be used in Apis mellifera research but not restricted to Nosema. Please, provide correct references.
L80- Please substitute “spreading” for “distribution”
L80-82. Please, remove this sentence. Nosemosis is usually used to the describe the disease and in this study, the authors analyzed the presence of N. apis and N. ceranae but this was not related with symptomatology. I would recommend restricting this word to sentences in this context. Also, I agree that the information regarding the presence and distribution of these pathogens is important and contribute to gain insights and compare it with other countries. However, this sentence is too ambitious.
L85-87- Which was the criteria of the number of samples taken from each region? Was it according to the number of colonies from each region? Please, provide details of the experimental design.
L91-92- Why were the samples taken from the entrance or from the frames away from the brood nest? Was any bias in the method of sample collection according with the region?
L95- Which cell lysis buffer? Please, provide information and references.
L125- Did the authors include positive controls? Did the authors verify the identity of at least one of the PCR fragments of each Nosema species? According to Martín-Hernández et al., 2007 the annealing of these primers is 61.8ºC and not 50ºC.
L128-remove specific
L135. Please, remove this sentence. It is already mentioned in L88
L137- Provide the percentage in order to compare with the infection level of the other two regions.
L141- Please, substitute “invasion” for “infection”. Also, “in comparison”
L153-Does the authors has historical information about the presence of N. apis and N. ceranae in the country?
L161- It has been also reported that there are bees tolerant and resistant to Nosema (Fontbonne et al., 2013) which is in agreement with your hypothesis. Authors may include this study in the discussion.
L162- I would recommend reorganizing the information in this paragraph. From the previous one, the authors point out that there might be bees more susceptible to Nosema than others and then argued that this may be the case in Bulgaria. But the first sentence only says that there are differences in the prevalence of Nosema in the flat and mountains areas. So, for a better understanding, it is important to say directly that these results might explain the differences in the prevalence observed in the different areas since different subspecies of bees are reared in each area.
L175-L177- Does the author has analyzed the genetic origin of the bees? If the authors still have the samples which were used for analyzing Nosema, it would be interesting to analyze their genetic origin and prove (or not) the hypothesis that they are proposing. I think that this analysis would contribute to making the article much more interesting.
L178- honey bee colonies
L182- Is this the case of the different regions in Bulgaria? Please, expand this idea.
L185- As I mentioned before, the authors analyzed the presence of N. apis and N. ceranae, so I would recommend using the word nosemosis only when talking about the disease.
L186-191- This information is not the conclusion. Please, remove it.
Table I- Please, re-write the title of the table. “Differential diagnostic investigations” is not appropriate.
Table I- “co-infection” instead of “co-invasion”
Fig. 1- I recommend to remove this figure.

·

Basic reporting

This manuscript reports on the presence of N. ceranae, an emerging exotic pathogen of the honey bee, in populations of honey bees in Bulgaria. The introduction and background top the study are well laid out, with some interesting references to the country’s beekeeping. Results are clear: N. ceranae is widespread, but the native congener Nosema apis was not detected.

The language of the ms is generally good throughout. I make some suggested changes in the section ‘general comments’ below, plus a suggestion for the inclusion of an additional reference. The structure of the ms is fine. Table 1 is clear, Figure 1 is of good quality but country labels should be added.

Experimental design

This ms clearly presents original research: results from PCR testing of honey bees from Bulgaria. The question is clearly posed, and the methods have been described in sufficient detail to allow replication of the study. On the technical side, the methods employed are generally robust. Below I list three additional points.

1. Were positive controls employed in PCRS to avoid the issue of false negatives? Duplicate PCRs might help in clarifying this issue if positive controls were not available.

2. Statistical testing needs to be clarified.

3. (line 99, and more a comment for future reference) Do not use EDTA in re-suspension buffer when dissolving DNA as it inhibits PCR (it chelates Mg ions of the PCR buffer). Use Tris alone.

4. (line 91) Were all 5 bees per colony analysed individually or as a single DNA extract?

Validity of the findings

In large part, the findings seem clear-cut and robust and sound.

1. (line 91) Why only 5 bees collected per colony when it is standard (Fries et al. 2013) to analyse 30 honey bees per colony?

2. Lines 132-142: Provide results of statistical tests to support statements

3. Lines 178-182 The authors hypothesise that warmer climates may favour N. ceranae over N. apis and vice versa. One paper explicitly exploring this issue is:
Natsopoulou et al. (2015) Interspecific competition in honeybee intracellular gut parasites is asymmetric and favours the spread of an emerging infectious disease. Proceedings of the Royal Society of London B: Biological Sciences 282:20141896.

Comments for the author

Suggested changes to the text (where relevant, by line number):

Abstract and elsewhere: Do not write about ‘Bulgarian honey bees’ as though they are different in some meaningful way from those in surrounding countries; rather, use ‘honey bees in Bulgaria’.
29-30 highest rate of invasion → highest prevalence
43 Give a date for the First Bulgarian Empire
48 Delete ‘etc’
49 various specific → numerous
53-4 Two main species of Nosema have been identified “in honey bees” (text in “” is needed)
66 replace ‘, but’ with a semicolon
69 invasion → infection
80 spreading → distribution
92 at entrance’ → at the entrance
94 add ‘a’ before ‘single bee’
119 add ‘a’ before ‘Little Genius’
122 add ‘a’ before ‘2%’
141 have demonstrated a higher level of invasion → had a higher prevalence of N. ceranae
145-7 Information should be transferred to the Introduction
165 infection → prevalence

·

Basic reporting

i recommended this article to published in Peer J

Experimental design

Experimental design is good, well defined

Validity of the findings

as this is the first report from Bulgaria, so i approved this manuscript to published

Comments for the author

In the submitted manuscript the authors present a study on the prevalence of Nosema ceranae and Nosema apis in honey bees in Bulgaria. However, I have several issues with the manuscript in its current form.
Since N. ceranae was first detected in 2005 in Apis mellifera colonies in Spain, many publications came up which confirm the presence of N.ceranae in many countries all over the world. It is now accepted that N.ceranae is distributed worldwide and it is assumed that it is replacing N.apis in several regions for so far not known reasons. It is thought that warmer climate conditions forward N. ceranae infections. Therefore, it is not stunning that N. ceranae can be detected in a Apis mellifera subspecies in Saudi-Arabia.
The submitted manuscript is yet another study on the prevalence of N. ceranae in yet another country and this is why the study is of low scientific interest.

my some minor comments are.

1. some latest references should be added in the text for instance "Ansari, M.J., Al-Ghamdi, A., Nuru, A., Khan, K.A. and Alattal, Y., 2017. Geographical distribution and molecular detection of Nosema ceranae from indigenous honey bees of Saudi Arabia. Saudi Journal of Biological Sciences.Volume 24, Issue 5, July 2017, Pages 983-991"

2. it will be more appropriate if author can add microscopic studies of nosema spores in the text

3. discussion should be elaborated in context to the prevalence of nosema in the adjacent countries of Bulgaria

---

## Round 0.2 · Minor Revisions

· Academic Editor

Minor Revisions

I think that the manuscript has been considerably improved but needs further minor revisions prior to publication.

Reviewer 1 ·

Basic reporting

In this study, Hristov et al. aimed to analyze the presence of Nosema apis and Nosema ceranae in samples taken from different regions of Bulgaria. The manuscript improved a lot since the last revision. I consider that it should be published now.

Experimental design

The experimental design is clear now and the required information was provided by the authors.

Validity of the findings

No comment

Comments for the author

Specific comments:
L39-46- Please, provide references.
L43- “On the other hand, honey bees are…”
L44- Remove “are”
L143-144- Please, submit the sequences to NCBI and provide their accession number

---

## Round 0.3 · Minor Revisions

· Academic Editor

Minor Revisions

Dear Pete,

Please could you carefully check your manuscript and resubmit it? I personally found many typos (e.g. Inroduction instead of introduction) and some unclear sentences (e.g. "On the other hand honey bees a valuable economic asset due to the ensemble of their products which are includes honey...").

All I'm kindly asking for is a little extra effort, but necessary in order to make your article wholly clear and publishable on PeerJ.

Thank you.

---

## Round 0.4 · accepted · Accept

· Academic Editor

Accept

The manuscript is now ready to be published.